# Thermal Characterisation and Isoconversional Kinetic Analysis of Osmotically Dried Pork Meat Proteins *Longissimus dorsi*

**DOI:** 10.3390/foods12152867

**Published:** 2023-07-27

**Authors:** Sanja Ostojić, Darko Micić, Snežana Zlatanović, Biljana Lončar, Vladimir Filipović, Lato Pezo

**Affiliations:** 1Institute of General and Physical Chemistry, Studentski trg 12/V, 11000 Belgrade, Serbia; micic83@gmail.com (D.M.); snezana.zlatanovic@gmail.com (S.Z.); latopezo@yahoo.co.uk (L.P.); 2Faculty of Technology, Novi Sad University of Novi Sad, Boulevard cara Lazara 1, 21102 Novi Sad, Serbia; biljanacurcicc@gmail.com (B.L.); vladaf@uns.ac.rs (V.F.)

**Keywords:** meat protein, denaturation, DSC, kinetic analysis

## Abstract

The kinetic properties and thermal characteristics of fresh pork meat proteins (*Longissimus dorsi*), as well as osmotically dehydrated meat proteins, were investigated using differential scanning calorimetry. Two isoconversional kinetical methods, namely the differential Friedman and integral Ortega methods, were employed to analyze the data. The obtained kinetic triplet, activation energy, pre-exponential factor, and extent of conversion, has been discussed. The resulting activation energy for proteins of fresh meat ranges between 751 kJ·mol^−1^ for myosin, 152 kJ·mol^−1^ for collagen and sarcoplasmic proteins, and 331 kJ·mol^−1^ for actin at a conversion degree of 0.1 to 0.9. For osmotically dried pork meat proteins, the values range from 307 kJ·mol^−1^ for myosin 272 kJ·mol^−1^ for collagen and sarcoplasmic proteins, and 334.83 kJ·mol^−1^ for actin at a conversion degree from 0.1 to 0.9. The proteins of the dry meat obtained by osmotic dehydration in molasses could be described as partly unfolded as they retain the characteristic protein denaturation transition. Concerning the decrease in enthalpies of proteins denaturation, thermodynamic destabilization of dried meat proteins occurred. On the contrary, dried meat proteins were thermally stabilized with respect to increase in the temperatures of denaturation. Knowledge of the nature of meat protein denaturation of each kind of meat product is one of the necessary tools for developing the technology of meat product processing and to achieve desired quality and nutritional value. The kinetic analysis of meat protein denaturation is appropriate because protein denaturation gives rise to changes in meat texture during processing and directly affects the quality of product.

## 1. Introduction

Food drying is a process of removing water content from food products, which inhibits the growth of microorganisms that cause spoilage and extends the shelf life of food. This preservation method has been used for thousands of years, and it remains an important technique for preserving food [1]. Meat has been present in human diet for more than 3 million years [2]. Based on their digestive system, humans are classified as omnivores, falling between their frugivorous anthropoid relatives (e.g., chimpanzees) and true carnivores [2]. Animal meat is one of the major high-quality protein sources in human nutrition. Rapid economic development is accompanied by increasing demand for meat, which puts pressure on the global meat supply [3]. Meat drying is a traditional method of preserving meat, and it has been shown to have significant nutritional benefits [4]. The process of meat drying removes water, which inhibits the growth of microorganisms that can cause spoilage. Reducing the water content in meat helps in preventing microbial growth, which in turn prevents the presence of harmful substances in meat [4]. One of the primary nutritional benefits of meat drying is that it can increase the protein content of the meat. As water is removed from the meat, the protein content becomes more concentrated, which can be beneficial for individuals who require a high-protein diet, such as athletes or those with certain medical conditions [4].

The process of osmotic dehydration represents a highly effective way of partially reducing the water content of the raw material, causing minimal damage to the quality of the final product [1]. This process can be used to produce products that are nutritionally enriched with an osmotic solution, in this study, sugar beet molasses, and the resulting products can be considered as functional foods [5]. Based on numerous studies, sugar beet molasses, a concentrated multicomponent system that is a by-product of the sugar industry, can be used as a highly suitable hypertonic solution [6]. Molasses has a very complex chemical composition. It is known that over 200 different compounds are present in molasses [6]. It contains over 80% dry matter, around 50% sucrose, 1% raffinose, 0.25% invert sugars, around 5% protein, around 1.5% purine and pyrimidine bases, some organic acids, and pectin. It contains B-group vitamins and minerals, with significant amounts of potassium, sodium, and iron [6,7,8]. Due to various phytochemical compounds, including phenols, plant sterols, flavonoids, and policosanols, molasses have gained interest due to their antioxidant activity, cholesterol-lowering properties, and other potential health benefits [9]. The composition of molasses is variable and primarily depends on the compounds present in the raw sugar beet, although the composition of molasses also depends on the way of carrying out the technological processes of sugar beet cleaning and the sugar crystallization process [6]. The greatest differences in the chemical composition relate to the quantities of raffinose, invert sugar, and mineral substances [6]. The significance lies in the fact that all the mineral matter is in the dissolved state due to the osmotic dehydration process. Molasses has humectant properties and affects the water activity of the product. Due to the constant immersion of the sample in the osmotic solution, the tissue is not exposed to oxygen, and, therefore, there is no need to use means to protect against oxidative and enzymatic changes. Research by Lončar et al. [6] has demonstrated that sugar beet molasses is an extremely effective medium for the osmotic dehydration of fruits, vegetables, and meat. The high dry matter content, specific composition of nutrients, low cost, and energy requirements are the main reasons why sugar beet molasses is such a useful osmotic solution [10].

The denaturation of meat proteins by heat is a complex process involving the disruption of weak intermolecular forces, such as hydrogen bonds, hydrophobic interactions, and van der Waals forces. This causes the proteins to lose their native conformation and unfold into a more extended conformation. The unfolded proteins then aggregate and form cross-links with each other, leading to the formation of a three-dimensional network that traps water and gives meat its characteristic texture. Apart from heat, other factors, such as pH, salt, and mechanical stress, can also cause protein denaturation in meat. Acidic conditions can lead to the denaturation of proteins by disrupting the ionic bonds that hold the protein together. High salt concentrations can cause the proteins to unfold by disrupting the hydrophobic interactions between protein molecules. Mechanical stress can also cause protein denaturation by physically disrupting the protein structure [11,12]. The conformational changes in proteins that occur during heating are commonly referred to as protein denaturation. The temperatures at which these changes occur are called protein denaturation temperatures. Protein denaturation has been extensively studied by differential scanning calorimetry [13]. Protein denaturation is a process where the three-dimensional structure of a protein is disrupted, leading to the loss of its biological activity. In meat, protein denaturation occurs due to various factors, such as heat, acid, salt, and mechanical stress during processing, cooking, and storage. One of the main causes of protein denaturation in meat is heat. Heating meat causes the proteins to denature and coagulate, leading to changes in the texture, flavor, and color of the meat. The denaturation of meat proteins begins at temperatures as low as 40 °C, but significant changes occur at temperatures above 60 °C. At higher temperatures, the proteins become more extensively denatured, leading to the formation of a gel-like structure, which is responsible for the firmness and texture of cooked meat [11,12].

This study presented the formation of a new protein matrix in dehydrated meat through the interactions between molasses components and meat proteins during osmotic dehydration. These interactions involve complex processes that cause changes in the structure of meat proteins, leading to the development of a dry meat protein which is enriched with molasses components and, thus, can be less susceptible to microbiological spoilage and, at the same time, enriched with high-value nutrients. In the literature, there are some data concerning the thermal properties of fresh meat proteins [14,15,16], but not as much about thermal properties of dehydrated meat proteins [17]. Furthermore, there is a lack of data in the literature on the thermal properties of osmotically dehydrated pork meat proteins and data about the kinetic behavior of osmotically dehydrated (in sugar beet molasses) pork meat proteins.

Several studies have conducted kinetic analysis of the changes that occur in protein-based foods during thermal processing. These studies have evaluated the changes in the texture of meat during heating weight-loss from the meat during cooking, color changes in fish during grilling, and fluctuations in the umami component of meat during cooking [14]. Based on the findings of these studies, empirical kinetic equations describing texture and weight loss have been generated from experimental values. These empirical equations are excellent for practical use, but their application is limited because the kinetic parameters may depend on heating conditions applied [14]. Therefore, a kinetic model based on the physical, chemical, or biochemical reactions leading to the target phenomenon is desirable for generating predictions corresponding to various heating conditions. The kinetic analysis of protein denaturation is appropriate because protein denaturation gives rise to changes in meat texture and weight loss during thermal processing [14]. No studies have performed a kinetic analysis of the changes that occur in meat considering the state of the protein denaturation during the process of osmotic dehydration. Furthermore, no research has been conducted on the kinetic analysis of protein denaturation in dried meat, and no study has discussed the state of protein denaturation in the dried meat. 

In this study, the kinetic properties and thermal characteristics of fresh and osmotically dehydrated pork meat (*Longissimus dorsi*) proteins were examined using two kinetical models: the isoconversional differential Friedman method [18,19,20] and an integral method for isoconversional data by Ortega [21]. The obtained kinetic triplet (E-activation energy, A-pre-exponential factor and f (α) extent of conversion) has been discussed. The aim of this work is to thermally characterize (by means of differential scanning calorimetry, DSC) and compare the fresh and dried meat proteins denaturation by obtaining the kinetic and thermodynamic parameters of the denaturation process.

## 2. Materials and Methods

### 2.1. Materials

Fresh pork meat (*Longissimus dorsi*), 48 h postmortem, was purchased from the butcher shop in Novi Sad and cleaned to remove visible fat and connective tissue before being cut into (1 × 1 × 1) cm cubes. Sugar beet molasses were obtained from a sugar factory (Crvenka, Serbia).

### 2.2. Osmotic Dehydration of Pork Meat in Sugar Beet Molasses

The osmotic dehydration procedure was performed at the Faculty of Technology, University of Novi Sad, according to the procedure described in the work of Šuput et al. [5]. The experimental pilot plant for discontinuous osmotic dehydration, which was designed and built at the same institution for the purpose of the scientific project supported by the Ministry of Education, Science, and Technological Development of the Republic of Serbia (TR20112, 2008), was utilized. Osmotic dehydration of fresh pork meat (*Longissimus dorsi*) was performed using sugar beet molasses. Fresh pork meat (*Longissimus dorsi*), 48 h postmortem, was cleaned to remove external fat and connective tissue and then cut to small (1 × 1 × 1) cm cubes. Then osmotic dehydration was performed using sugar beet molasses. The average dry matter content of molasses, determined by refractometry, was 84.54%. To achieve the desired concentrations of the osmotic solution, molasses was diluted with distilled water. Dehydration was performed at a temperature of 22 °C for 5 h. The ratio of osmotic solution to meat was 5:1 (*w*/*w*). After the processing time, the meat samples were lightly washed with distilled water and gently blotted to remove excessive water from the surface. After the osmotic dehydration, dried meat was packed in foil (transparent multi-layered foil PVC/PE-EVOH-PE: polyvinyl chloride/polyethylene-ethylene-vinyl alcohol-polyethylene) under 30% CO_2_ + 70% N_2_ gas [5]. 

### 2.3. Determination of Proximate Composition and Water Activity (a_w_)

The chemical composition of *Longissimus dorsi* (moisture, protein, ash, and fat) was determined by standard methods for meat and meat products. The water content (dry matter) was determined following the ISO 1442:1998 standard [22]; the determination of protein content was according to the standard ISO 1871:1992 [23], Agricultural and Food Products—General Guidance for Nitrogen Determination by Kjeldahl Method; the determination of total ash content was based on standard ISO 936:1999 [24], the determination of total fat content was according to standard ISO 1443:1992 [25]. The water activity (a_w_) was determined by a_w_ meter Novasina Lab Swift-aw, Novasina AG, Lachen, Switzerland, at 25 °C.

### 2.4. Thermal Analysis 

DSC experiments of fresh and osmotically dehydrated meat samples were performed on a differential scanning calorimeter DSC Q 1000 instrument, with an RCS cooling system from TA Instruments, New Castle, Delaware, USA. The instrument was calibrated for temperature and enthalpy according to the standard instructions of the manufacturer using standard metal indium whose melting point temperature is T = 156.59 °C, and whose melting enthalpy ΔH = 28.18 Jg^−1^ [26]. All experiments were conducted in the nitrogen flow (purity stream of 99.999%) at a gas flow rate of 50 mL min^−1^ in a DSC cell. The samples of fresh and osmotically dehydrated meat were weighed to the mass 9–13 mg by the analytical scale balance from Mettler Toledo AE 163, Toledo. The samples were analyzed in hermetically sealed aluminium pans manufactured by TA Instruments in the temperature range from 5 °C to 120 °C. To check the reversibility of protein denaturation, all scanned samples where rescanned after cooling. The hermetic closure of the pans was achieved using a blue press from the manufacturer TA Instruments.

Each thermogram was analyzed using TA Advantage Universal analysis 2000 software to obtain the meat protein denaturation temperatures (onset, T_onset_; midpoint, T_d_; final T_end_) and enthalpy ΔH_d_ of meat protein denaturation. 

#### Kinetics of Meat Protein Thermal Denaturation

To determine the kinetic parameters of thermal denaturation of the meat proteins, the samples were hermetically sealed in hermetic aluminum pans and scanned. The approach employed for conducting the kinetic studies followed the recommendations for collecting kinetic data [27] and for performing kinetic computations [28] outlined by the ICTAC Kinetics Committee.

The activation energy (*E_a_*) of the protein denaturation process in both fresh and dried meat samples were determined using two isoconversional non-isothermal methods. The first method applied was the differential method developed by Friedman (Equation (1)) [20], while the second method employed was the integral method developed by Ortega (Equation (2)) [21]. The following equations were used:(1)$$\ln\left[\beta_{i}{\left(\frac{d\alpha }{dT}\right)}_{\alpha ,i}\right]=const-\frac{E_{\alpha }}{RT_{\alpha ,i}}$$
(2)$$\ln\left(\frac{\beta_{i}}{\Delta T_{\alpha ,i}}\right)=const-\frac{E_{\alpha }}{RT_{\alpha ,i}}$$
where *β* is the heating rate (Kmin^−1^), *α* is extent of conversion (the area under the peak up to a given temperature is proportional to the extent of conversion), *R* is the universal gas constant (8.314 J (mol K) ^−1^) and *T* is temperature (K). The indexes *i* and *α* correspond to a specific heating rate and the extent of conversion, respectively. In Equation (2) Δ*T_α_* is *T_α_* − *T*_*α*−Δ*α*_, where Δ*α* = 0.02. For each specific *α*, the value of *E_α_* is calculated from the slope of a linear regression of the left side of Equations (1) and (2) against 1/*T_α,i_* (where *T_α,i_* represents the temperature at which the extent of conversion *α* is achieved under the *i*-th heating rate). The samples of fresh and dried meat were heated at six different heating rates, *β_i_* (0.5; 1.5; 2; 3; 5 and 10 °Cmin^−1^), starting from 5 to 120 °C, and *E_α_* values were calculated in a range of *α* from 0.05 to 0.95 with a step of 0.05 for both methods. The logarithmic values of the pre-exponential factors at the extent of conversion *α* (ln(A*α*)) were determined utilizing the compensation effect as described by Vyazovkin [29]. Based on *E*, A, and *T* values on the corresponding extent of conversion *α* = 0.5, the rate constants (k_0.5_) of the protein denaturation in fresh and dried meat were calculated by the Arrhenius equation.

### 2.5. Statistics

All measurements were performed in triplicate, and the results were expressed as means ± SD. XLSTAT (version 2014.5.03, Addinsoft, New York, NY, USA) analysis and the statistics add-in for MS Excel were used for statistical analysis. The obtained values were subjected to a three-way ANOVA. To compare the means of these parameters, a post hoc Tukey’s test was used (*p* < 0.05).

## 3. Results

### 3.1. Proximate Composition of Pork Meat (Longissimus dorsi) and Sugar Beet Molasses

The results of proximate analysis of fresh pork meat *(Longissimus dorsi*) used, in this study, for osmotic dehydration were as follows: moisture 75.6 ± 2.3%, protein 21.7 ± 2.2%, total fat 1.3 ± 0.3%, and total ash 1.2 ± 0.2%. For the dried pork meat, the composition was as follows: moisture 37.7 ± 3.5%, protein 54.6 ± 6.1%, fat 0.9 ± 0.2%, total ash 6.8 ± 0.5%. The chemical composition of sugar beet molasses applied in the research was as follows: sucrose 49.8 ± 2.2%, total reducing sugars 52.2 ± 3.3%, invert sugars 0.49 ± 0.1%. Water activity (a_w_) for fresh *Longissimus dorsi* obtained for fresh pork meat was a_w_ = 0.960, while for dried pork meat a_w_ = 0.729 at 25 °C.

### 3.2. Thermal Analysis of Pork Meat Proteins Longissimus dorsi

On DSC curves of fresh and dehydrated pork meat samples three endothermal thermal transitions can be recognized, which correspond to the thermal denaturation of meat proteins. According to the literature data, three peaks of denaturation are assigned to myosin, collagen sarcoplasmic protein, and actin. The first transition with a temperature maximum between 54–58 °C represents the thermal denaturation of myosin (P1) [30,31,32,33]. The second transition, which has a temperature maximum between 65 and 67 °C, corresponds to the thermal denaturation of collagen [19,34,35] and sarcoplasmic proteins (P2) [11]. The third transition is the thermal denaturation of actin, which is between the temperatures of 80 °C and 83 °C (P3) [36]. For the second transition, (P2) it was shown that both the isolated proteins actomyosin and myosin and their subunits have thermal denaturation in the same temperature range [36].

DSC curves obtained for fresh meat at different heating rates are shown in Figure 1. In Table 1, the thermodynamic parameters of proteins denaturation of fresh and dried meat samples at different heating rates are given. It can be observed that the meat drying process influenced both denaturation temperatures (T_do_ and T_dp_) of all three proteins in dried meat, resulting in higher values compared to fresh meat at all heating rates (*p* < 0.05). When it comes to denaturation enthalpy, the meat drying process had no significant effect on its values for all three proteins at all heating rates (*p* > 0.05).

From the obtained results, there is an evident shift in temperature as well as changes in the enthalpy (ΔH_d_) of denaturation of proteins at different heating rates for proteins of the fresh pork meat and for the transitions obtained for proteins of osmotically dried pork meat.

All samples of fresh and dried meat proteins analyzed by DSC did not show reversibility of protein denaturation after rescanning. This means that the protein denaturation of studied fresh and dried meat proteins occurred irreversibly.

#### Kinetic Analysis

The kinetic parameters of meat protein denaturation were obtained using a non-isothermal kinetic model used by Istrate et al. [18], who applied a non-isothermal kinetic model to characterize the denaturation kinetics of human hair.

The dependence of activation energy on the extent of conversion (α) with the Friedman and Ortega isoconversion methods [18,19,20,21] was obtained. Data for calculation of the kinetic parameters were obtained from a series of DSC experiments performed at different heating rates, but with the same extent of conversion. The dependence of activation energy on the conversion rate for thermal transitions corresponding to the denaturation of fresh and dried meat proteins myosin (Protein 1 (P1)), collagen/sarcoplasmic protein (Protein 2 (P2)), and actin (Protein 3 (P3)), are shown on Figure 2.

The kinetic triplet (E, lnA, and k_0.5_) of the thermal denaturation process of proteins in fresh and dried meat, obtained using the differential Friedman and integral Ortega methods, are listed in Table 2. Based on the results of the three-way ANOVA, it can be concluded that there was no statistically significant difference between the applied methods (*p* > 0.05). For fresh meat, P1 exhibited the highest value of activation energy, while P2 had the lowest (*p* < 0.05). In the case of dried meat, there was no statistically significant difference in the values of activation energy among the proteins (*p* > 0.05), indicating that the activation energy was uniform among all proteins. Regarding the rate constants at a conversion degree of 0.5 (k_0.5_), it can be observed that the drying of meat resulted in a decrease in the value of k_0.5_ for P1 and P3, while there was a slight increase in the value of k_0.5_ for P2 in the comparison to the fresh meat proteins.

It has been shown that the E obtained for fresh meat protein P1 increased with the extent of α, while for proteins P2 and P3, it slightly decreased or remained at about the same value (P2) when increasing the extent of conversion (α) (Figure 2a). The obtained E (Ortega and Friedman) for proteins of fresh pork meat ranged from 750 kJ·mol^−1^ for myosin (P1), to 152 kJ·mol^−1^ for collagen and sarcoplasmic proteins (P2), and 334 kJ·mol^−1^ for actin (P3) at a conversion degree (α) from 0.1 to 0.9. (Table 2, Figure 2a). Generally, the activation energies in terms of the extent of conversion (α) from 0.1 to 0.9, obtained for the denaturation of proteins P1, P2 and P3 of the fresh meat, were in the range from 100 to 1400 kJ·mol^−1^ (Figure 2a).

The obtained E (Ortega and Friedman) for proteins of osmotically dried pork meat ranged from 307 kJ·mol^−1^ for myosin (P1), to 272 kJ·mol^−1^ for collagen and sarcoplasmic proteins (P2), and 334 kJ·mol^−1^ for actin (P3) at the conversion degree (α) from 0.1 to 0.9. (Table 2, Figure 2b). E, in the function of α from 0.1 to 0.9, obtained for the denaturation of proteins P1, P2 and P3 of osmotically dried meat was in the range from 200 to 450 kJ·mol^−1^ (Figure 2b). It can be seen that the values of E obtained for the proteins of osmotically dried meat are lower than the E values found for fresh meat.

## 4. Discussion

Ishiwatari and the authors in their work [14] have shown the importance of the kinetic analysis of the denaturation of meat proteins myosin and actin. It has been shown that with the obtained kinetic data of protein denaturation, it is possible to predict the degree of denaturation of the meat proteins. Same authors showed that there is a significant influence of actin denaturation on the meat elastic modulus, while the effect of denaturation of myosin is negligible. However, in the early phase of denaturation of myosin, it was shown that there was a change in the state of water. Thus, the existence of a direct influence on the state of denaturation of protein on the texture and the state of water in the meat system has been confirmed [14].

On DSC curves of fresh and dehydrated pork meat, three endothermal thermal transitions can be recognized, which correspond to the thermal denaturation of meat proteins (Table 1 and Figure 1). According to the literature data, the three peaks of denaturation are assigned to myosin (P1), collagen and sarcoplasmic protein (P2), and actin (P). The first transition with a temperature maximum between 54–58 °C represents the thermal denaturation of myosin [30,31,32,33,37]. The second transition, which has a temperature maximum between 65 and 67 °C, corresponds to the thermal denaturation of collagen [34,35] and sarcoplasmic proteins [11]. The third transition is the thermal denaturation of actin, which occurs between temperatures of 80 and 83 °C [36]. For the second transition, it was shown that both the isolated proteins actomyosin and myosin and their subunits experience thermal denaturation in the same temperature range [36].

Obtained activation energies (Table 2) are in agreement with the literature, as Kajitani et al. [37] studied thermal denaturation of protein in cured pork meat, and found the averaged activation energy of each protein (myosin: 2.41 × 10^2^ kJ·mol^−1^, sarcoplasmic proteins and collagen: 3.26 × 10^2^ kJ·mol^−1^, actin: 2.50 × 10^2^ kJ·mol^−1^).

It has been shown that activation energies obtained for fresh meat protein P1 increased with the extent of conversion, while for proteins P2 and P3 they remain at about the same value (P2) (Figure 2a), with no statistical significance. Increasing the activation energy with conversion can be widely applied for the thermal decomposition of many polymers through competing, consecutive, and some independent reactions [18]. The slight decrease in conversion finds for proteins P2 and P3 corresponds to the kinetic scheme of endothermic reversible reactions [18]. Generally, it is considered that any changes in E with conversion are characteristic of the complex processes, in this case protein denaturation. This means that complex competitive reaction sequences or those complicated by diffusion E vary with α [18]. Complex processes of protein denaturation take place simultaneously, whether involving the denaturation of one protein (protein domains) or the simultaneous denaturation of several proteins, as in this study is the case with the endothermal transition P2, which is considered to be the denaturation of collagen [19,34,35,37] and sarcoplasmic proteins [11]. Protein P1 myosin denaturation is also a complex process, including the myosin subfragment denaturation [33].

The resulting E (Ortega and Friedman) for proteins of fresh meat ranges between 751 kJ·mol^−1^ for myosin (P1), 152 kJ·mol^−1^ for collagen and sarcoplasmic proteins, and 331 kJ·mol^−1^ for actin (P3) at a conversion degree of 0.1 to 0.9 (Table 2, Figure 2a). The obtained results for E (Ortega and Friedman) are in agreement with the literature data [18,19] and correspond to the wide range of E values obtained for the thermal denaturation of mammalian tissue [19,38].

Vyazovkin, in [19], showed that results obtained for the dependence of E and α are obviously due to a higher degree of the denaturation process. The obtained E (Ortega and Friedman) for proteins of osmotically dried pork meat range from 307 kJ·mol^−1^ for myosin (P1), 272 kJ·mol^−1^ for collagen and sarcoplasmic proteins (P2), and 280 kJ·mol^−1^ for actin (P3) at a conversion degree from 0.1 to 0.9. (Table 2, Figure 2b).

The activation energy values obtained for the proteins of osmotically dried meat are lower than the E values obtained for fresh meat, except for protein P2 (collagen and sarcoplasmic proteins). This discrepancy can be explained by the fact that during osmotic dehydration, the interaction with the components of the osmotic solution and protein–protein interactions took place, which led to the construction of a new structure among the P2 proteins; a consequence of this is an increase in E compared to the P2 proteins of fresh meat. Generally, those findings are in agreement with the first scenario of protein kinetic stability proposed by Sanchez-Ruiz [13]. Therefore, it can be assumed that the newly formed protein matrix is thermally stabilized but thermodynamically and kinetically less stable in comparison to the fresh meat. It is known that proteins can be thermally stabilized with interactions with various ligands and also that proteins can be stabilized by protein–protein association in solutions [39]. For proteins with a low moisture content there is not a lot of data in the literature concerning denaturation processes. Atuonwu et al. [40] have been studying the kinetics of whey protein denaturation under different moisture contents. They concluded that there is a dependence between the moisture content and rate constants of protein denaturation. Resulting rate constants were found to increase with both temperature and moisture content [40]. This is in agreement with the results found in this study, as rate constants of protein denaturation (k_0.5_) (Table 2) obtained for dried meat are mainly lower than rate constants of protein denaturation of fresh pork meat. Obtained values for rate constants were generally in agreement with the literature [37]. According to the results found in this study, rate constants (k_0.5_) of dried meat protein denaturation are lower in comparison to the rate constants of the denaturation of fresh pork meat proteins, indicating a kinetic stabilization of the dry meat protein matrix. An exception was for the denaturation of proteins corresponding to P2 (collagen and sarcoplasmic proteins) in fresh and dry meat whose rate constants were higher for dry meat proteins. Also, Kajitani et al. [37] studied the influence of the NaCl level on the kinetic constant of pork meat protein denaturation and temperature dependency using the DSC dynamic method. They found that as the level of NaCl in the meat increased, the thermal-denaturation rate constant of each protein increased, and the rate constant for actin especially increased, e.g., the rate constant at 70 °C increased from 0.1 min^−1^ (2 mg·g^−1^ of NaCl) to 1.75 min^−1^ (40 mg·g^−1^ of NaCl). Those results for rate constants are comparable to the results obtained in this study for the newly formed protein matrix (Table 2).

Atuonwu et al. [40] found a dependence between the moisture content and rate constants of protein denaturation, indicating that for each drying processing time, there is a family of temperature–moisture content combinations that achieve equal denaturation, i.e., raising the moisture content lowers the temperature required. For dried meat proteins obtained in this work, at a reduced moisture content of 37.70% and water activity of a_w_ = 0.729, it can be concluded that thermal stabilization (according to temperatures of denaturation of dried proteins shifting to the higher temperatures in comparison to denaturation temperatures of proteins of fresh meat) has been achieved through protein–protein association and interactions with components of the osmotic solution. Temperatures of denaturation (T_d (1,2,3)_) obtained for dried meat proteins in this work were at comparatively higher temperatures, which could be due to the initiation of mobility in protein molecules, what is in agreement with the literature [41,42]. Phan-Xuan et al. [43] also concluded in their study of lysozyme denaturation in solid and liquid states that, when decreasing the water content and water activity, the denaturation temperature shifts to higher values. This behavior illustrates very well the importance of water in the stability of proteins. Since water is needed for denaturation, dehydration makes the protein less susceptible to thermal degradation [43]. These conclusions are in agreement with the results obtained in this study. It can be concluded that the initiation of protein molecule mobility could be induced not only by water loss but also with protein interactions with molasses components. Obtained enthalpies of protein denaturation ΔH_d (1,2,3)_ (Table 1) are in agreement with the literature [44]. Also, it can be concluded that dried meat proteins are thermodynamically less stable according to the slightly lower enthalpies of dried meat protein denaturation ΔH_d (1,2,3)_ in comparison with proteins of fresh meat (Table 1). It can be proposed that the kinetic stability of dried meat proteins can be classified into scenario (1) [13], meaning that the native (N), functional protein (in this case fresh meat protein) is thermodynamically stable with respect to unfolded and partially-unfolded (U) states (in this work, the partly unfolded state obtained is a newly formed protein matrix).

However, these states can undergo irreversible alteration processes (aggregation, proteolysis, strong interactions with other macromolecules, interaction with components of the osmotic solution as in this study, etc.) that lead to some kind of “final state” (unable to fold back to the native one). This scenario is summarized in the Lumry–Eyring model: N⇔U⇒F, where F is the final state [13].

In this study, dried meat proteins could be described as partly unfolded or partly denatured, as they retain their characteristic denaturation endothermal transition of proteins (P1, P2, and P3). Concerning the decreased ΔH_d_ of the denaturation of proteins (P1, P2, P3), it can be proposed that thermodynamic destabilization of dried meat proteins occurred in comparison to the ΔH_d_ of denaturation obtained for proteins of fresh meat. On the contrary, dry meat proteins are thermally stabilized regarding the increase in the temperatures of denaturation (Table 1). It is well known that sugar solutions, especially sucrose (which is the most abundant carbohydrate present in molasses [6,8]) can stabilize the protein structure [45]. By now, the most studied phenomenon was a marked increase in the thermal and conformational stability of the globular proteins in an aqueous medium in the presence of sugar [46]. The mechanism of the action of sugar on protein was proposed to be that the sugar molecules are preferentially excluded from the region immediately surrounding the proteins [46]. Ferreira et al. [47] studied the effects of osmolytes (sucrose and trehalose) on protein–solvent interactions in crowded environments and found that proteins’ responses to the presence of different osmolytes are governed by the proteins’ structures. Cao et al. [48] gave a systematic analysis of protein–carbohydrate interactions and binding sites. Those findings, which considered the globular protein stabilization in aqueous medium by sugar molecules, can be applied to this work as well, since meat fibrillar proteins clearly interact with the osmotic solution (sugar cane molasses) and the dried proteins are thermally stabilized and partly unfolded (Table 1). According to Sanchez-Ruiz [13], modifications of the basic Lumry–Eyring model involving partially-unfolded states (in this case dry meat proteins) as they finally irreversibly denaturate into final state may be considered. In the type of situation described by a Lurmy–Eyring model, the protein will eventually end up in the non-functional, final state, as happened, in this study, to the dried meat proteins, as they irreversibly denatured into a final state (F) (Table 1). Therefore, as it has been stated by Sanches-Ruiz [13] that a biological function requires kinetic stabilization, i.e., a significant free energy barrier along the way from N to F. The Lumry–Eyring model is important because irreversible alterations of unfolded or partially unfolded states may be expected to occur efficiently in crowded and/or harsh, environments (in this study, molasses is used as an osmotic solution and concentrated protein environment). The Lumry– Eyring model suggests, therefore, that natural selection has endowed many proteins with kinetic stability, even if this fact is not revealed in the “traditional” in vitro experiments [13].

Quevedo et al. [49] studied whey protein denaturation at a high protein concentration (70%) and concluded that although the determined values of activation energy are in accordance with previous studies, it was found that the shear stress influenced the denaturation reaction of single and multi-component protein systems in different ways. The same authors, [49] showed that additional information about the denaturation mechanisms of whey proteins at high protein concentration could be gained by combining protein chemistry and rheological analysis. They concluded that changes in the concentration and ratio of different proteins of whey protein isolate could lead to changes in the denaturation and aggregation behavior of complex systems, such as whey protein isolate. Liu et al. [50] studied changes in the structures and properties of collagen fibers during collagen casing film manufacturing and found that dried collagen casing had the lowest enthalpy value and the lowest thermal denaturation temperature; this indicated that the thermal stability of the new structure was clearly inferior to the natural structure of the raw material, and similar protein destabilization has been found in this study.

It can be proposed that a similar influence of shear stress can be applied to the dried meat proteins obtained in this study. Further investigations are needed to fully understand the denaturation of dried meat proteins obtained in this study.

## 5. Conclusions

In the presented research, dried meat protein obtained through osmotic dehydration with sugar beet molasses was studied. The achieved thermal stabilization, through protein–protein association and interactions with the osmotic solution (molasses), resulted in the formation of dry meat protein. It was proposed that dried meat proteins are in a partly unfolded state and, regarding the kinetic stability, follow scenario (1), described previously. Therefore, it can be concluded that the osmotically dried meat proteins are thermally stabilized but thermodynamically and kinetically less stable in comparison to the fresh meat proteins.

Based on previous studies, it can be suggested that shear stress may also play a role in protein denaturation of the formed protein matrix. This research obtains deeper insights into the process of protein denaturation of the osmotically dried meat proteins and interactions between the meat proteins and osmotic solution used for dehydration, in this case sugar beet molasses, by usage of the DSC technique. It was demonstrated that with the simple usage of the DSC technique, it is possible to describe complex processes which take place during the osmotic dehydration of meat proteins, and that the DSC technique can be an excellent tool for following the changes that occur during the process of dehydration. Further investigations are needed to fully understand the denaturation of meat proteins osmotically dried in molasses.

## Figures and Tables

**Figure 1 foods-12-02867-f001:**
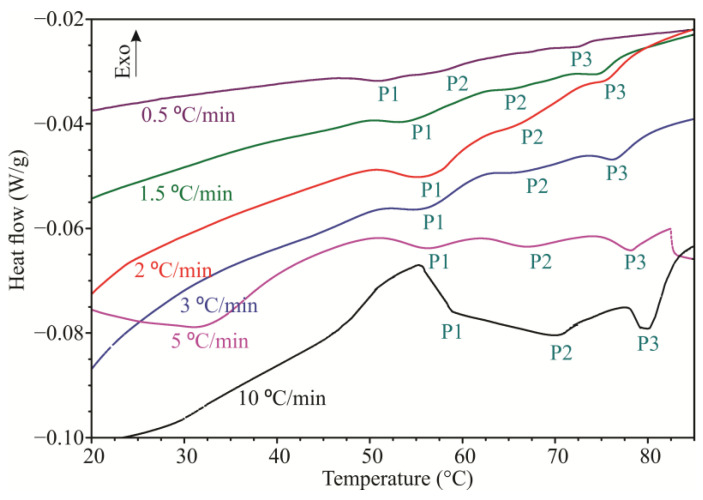
DSC curves of fresh meat protein denaturation obtained at different heating rates.

**Figure 2 foods-12-02867-f002:**
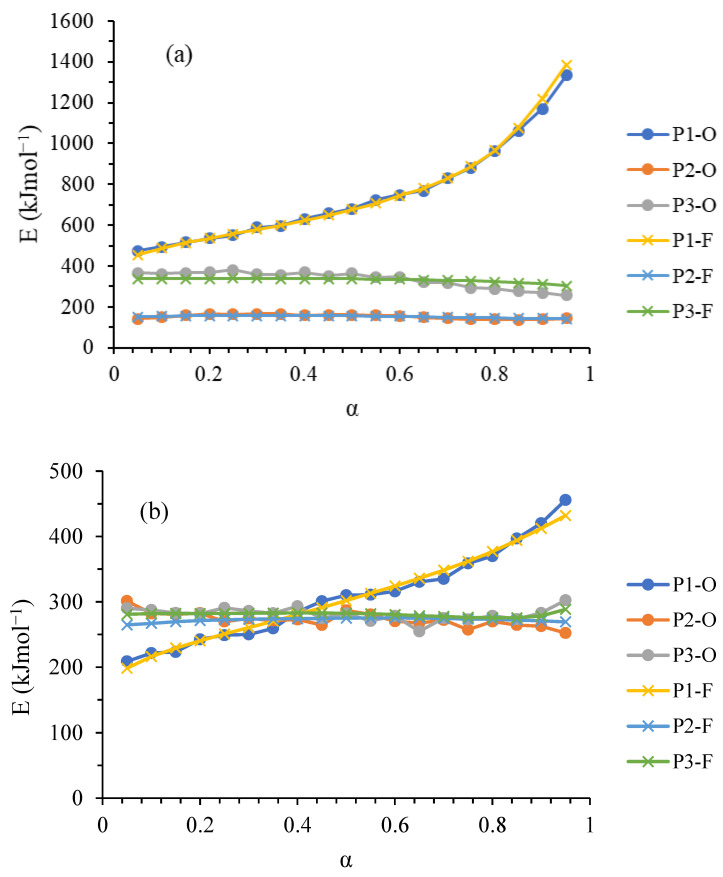
The dependence of the activation energy (E) from the conversion rate (α) for thermal transitions corresponding to the denaturation of the proteins of (**a**) the fresh and (**b**) the osmotically dried pork: myosin (P1), collagen/sarcoplasmic (P2), and actin (P3), O—Ortega kinetic model, F—Friedman kinetic model.

**Table 1 foods-12-02867-t001:** Thermodynamic parameters of fresh (FM) and dried (DM) meat proteins denaturation obtained at different heating rates (T_do (1,2,3)_—onset temperature of denaturation, T_dp (1,2,3)_—temperature of denaturation, ΔH_d (1,2,3)_—enthalpy of denaturation of proteins P1, P2, and P3).

Heat Rate (°Cmin^−1^)	P1			P2			P3		
	T_do1_ (°C)	T_dp1_ (°C)	ΔH_d1_ (Jg^−1^)	T_do2_ (°C)	T_dp2_ (°C)	ΔH_d2_ (Jg^−1^)	T_do3_ (°C)	T_dp3_ (°C)	ΔH_d3_ (Jg^−1^)
FM									
0.5	48.0 ± 0.0 ^A^	51.2 ± 0.1 ^A^	0.38 ± 0.01 ^A^	54.3 ± 0.1 ^B^	57.8 ± 0.3 ^B^	0.22 ± 0.03 ^A^	69.4 ± 0.2 ^B^	72.3 ± 0.1 ^B^	0.21 ± 0.02 ^A^
1.5	50.1 ± 0.4 ^B^	54.2 ± 0.1 ^B^	0.69 ± 0.20 ^A^	62.7 ± 0.9 ^B^	65.5 ± 0.2 ^B^	0.09 ± 0.06 ^A^	72.1 ± 0.6 ^B^	75.2 ± 0.6 ^B^	0.19 ± 0.04 ^A^
2	51.1 ± 0.1 ^B^	56.0 ± 0.3 ^B^	0.69 ± 0.14 ^A^	63.4 ± 0.4 ^B^	66.1 ± 0.7 ^B^	0.09 ± 0.08 ^A^	73.1 ± 0.2 ^B^	75.6 ± 0.0 ^B^	0.15 ± 0.02 ^A^
3	51.2 ± 1.0 ^B^	55.6 ± 1.3 ^B^	0.26 ± 0.12 ^A^	63.4 ± 0.1 ^B^	66.5 ± 0.8 ^B^	0.07 ± 0.04 ^A^	74.1 ± 0.4 ^B^	76.6 ± 0.2 ^B^	0.13 ± 0.06 ^A^
5	52.3 ± 0.1 ^B^	56.0 ± 0.3 ^B^	0.09 ± 0.03 ^A^	63.0 ± 0.1 ^B^	66.7 ± 0.8 ^B^	0.09 ± 0.05 ^A^	75.5 ± 0.6 ^B^	78.1 ± 0.2 ^B^	0.13 ± 0.06 ^A^
10	56.3 ± 0.2 ^B^	59.0 ± 0.7 ^B^	0.13 ± 0.09 ^A^	64.4 ± 2.1 ^B^	70.2 ± 0.3 ^B^	0.09 ± 0.04 ^A^	77.5 ± 0.8 ^B^	79.7 ± 0.6 ^B^	0.12 ± 0.02 ^A^
DM									
0.5	43.3 ± 0.5 ^B^	49.2 ± 0.2 ^B^	0.20 ± 0.02 ^A^	66.0 ± 0.0 ^A^	66.8 ± 0.1 ^A^	0.02 ± 0.01 ^A^	74.3 ± 0.3 ^A^	78.0 ± 0.3 ^A^	0.35 ± 0.02 ^A^
1.5	58.8 ± 0.1 ^A^	60.7 ± 0.3 ^A^	0.01 ± 0.00 ^A^	68.6 ± 0.2 ^A^	70.7 ± 0.2J ^A^	0.03 ± 0.00 ^A^	77.8 ± 0.3 ^A^	80.9 ± 0.3 ^A^	0.35 ± 0.01 ^A^
2	54.0 ± 1.5 ^A^	60.0 ± 0.3 ^A^	0.28 ± 0.14 ^A^	68.6 ± 0.2 ^A^	70.9 ± 0.0J ^A^	0.06 ± 0.04 ^A^	76.2 ± 1.1 ^A^	81.3 ± 0.9 ^A^	0.46 ± 1.16 ^A^
3	55.0 ± 1.5 ^A^	60.9 ± 0.5 ^A^	0.27 ± 0.21 ^A^	68.8 ± 0.8 ^A^	71.5 ± 1.6 ^A^	0.05 ± 0.04 ^A^	78.4 ± 1.1 ^A^	82.0 ± 0.4 ^A^	0.40 ± 0.11 ^A^
5	59.8 ± 0.7 ^A^	64.1 ± 0.2 ^A^	0.03 ± 0.01 ^A^	70.6 ± 0.7 ^A^	74.0 ± 0.5 ^A^	0.06 ± 0.04 ^A^	79.2 ± 1.6 ^A^	82.8 ± 0.5 ^A^	0.39 ± 0.16 ^A^
10	60.4 ± 2.5 ^A^	62.3 ± 1.1 ^A^	0.11 ± 0.18 ^A^	72.8 ± 0.6 ^A^	75.9 ± 0.5 ^A^	0.18 ± 0.11 ^A^	82.1 ± 0.5 ^A^	85.9 ± 0.5 ^A^	0.74 ± 0.15 ^A^

Data were subjected to three-way ANOVA (factors: proteins—three levels: P1, P2, and P3, degree of freedom was 2 and *p* < 0.05 for all parameters—T_do_, T_dp_, and ΔH_d_; treatment—two levels: FM and DM, degree of freedom was 1, *p* < 0.05 for T_do_ and T_dp_ and *p* > 0.05 for ΔH_d_; heat rate—six levels: 0.5, 1.5, 2, 3, 5, and 10, degree of freedom was 5 and *p* < 0.05 for all parameters—T_do_, T_dp_, and ΔH_d_; interaction “treatment × protein”; degree of freedom was 2 and *p* < 0.05 for all parameters; interaction “treatment × Heat rate”; degree of freedom was 5 and *p* < 0.05 for all parameters; interaction “protein × Heat rate”; degree of freedom was 10 and *p* < 0.05 for all parameters; interaction “treatment × protein × Heat rate”; degree of freedom was 10 and *p* < 0.05 for all parameters); different uppercase letters within the same column and same Heat rate indicate a significant difference of means, according to Tukey’s HSD test (*p* < 0.05).

**Table 2 foods-12-02867-t002:** Kinetic parameters of meat protein denaturation obtained from Freidman and Ortega kinetic models (FM—fresh meat, DM—dried meat), E_a_—mean value of activation energy at conversion degree (α) from 0.1 to 0.9; ln (A/min^−1^)—the natural logarithm of the prexponential factor; k_0.5_—protein denaturation rate constant at the conversion extent (α) value of 0.5.

			E (kJ·mol^−1^)	Ln (A/min^−1^)	k_0.5_ (min^−1^)
FM	Friedman	P1	751.1 ± 256.4 ^A^	273.6 ± 93.2 ^A^	3.86
		P2	152.1 ± 5.2 ^C^	53.6 ± 1.8 ^C^	1.26
		P3	331.2 ± 10.4 ^B^	113.8 ± 3.6 ^B^	2.85
	Ortega	P1	747.6 ± 240.7 ^A^	272.4 ± 87.5 ^A^	3.87
		P2	152.5 ± 10.9 ^C^	53.8 ± 3.8 ^C^	1.28
		P3	334.8 ± 39.5 ^B^	115.0 ± 13.5 ^B^	2.92
DM	Friedman	P1	307.2 ± 68.0 ^B^	110.2 ± 24.4 ^B^	1.54
		P2	272.4 ± 3.0 ^B^	94.6 ± 1.0 ^BC^	2.01
		P3	280.8 ± 3.2 ^B^	94.8 ± 1.1 ^BC^	1.91
	Ortega	P1	307.6 ± 70.2 ^B^	110.4 ± 25.2 ^B^	1.57
		P2	272.9 ± 11.2 ^B^	94.7 ± 3.9 ^BC^	2.04
		P3	281.1 ± 10.2 ^B^	94.9 ± 3.4 ^BC^	1.91

Data were subjected to three-way ANOVA (factors: proteins—three levels: P1, P2, and P3, degree of freedom was 2, F = 177.2 and *p* < 0.0001; treatment—two levels: SU and SV, degree of freedom was 1, F = 78.4 and *p* < 0.0001; model—two levels: Friedman and Ortega, degree of freedom was 1, F = 0.0004 and *p* = 0.984; interaction “model × protein”; degree of freedom was 2, F = 0.0053 and *p* = 0.994; interaction “model × treatment”; degree of freedom was 1, F = 0.000 and *p* = 0.995; interaction “protein × treatment”; degree of freedom was 2, F = 139.9 and *p* < 0.0001; interaction “model × protein × treatment”; degree of freedom was 2, F = 0.0054 and *p* = 0.995); different uppercase letters within the same column indicate a significant difference of means, according to Tukey’s HSD test (*p* < 0.05).

## Data Availability

Data are contained within the article.

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
