# Peer review of "Thermal Characterisation and Isoconversional Kinetic Analysis of Osmotically Dried Pork Meat Proteins Longissimus dorsi"

_foods, 2023, doi:10.3390/foods12152867_

Round 1

Reviewer 1 Report

Many researchers have used DSC for several decades to establish thermodynamic parameters of muscle meats. The authors of the manuscript entitled "Thermal characterization and kinetic analysis of osmotically dried pork meat Longissimus dorsi" studied the effect of molasses in the drying process of pork meat. Then, two kinetic models were implemented to extract the parameters defining the extent of protein denaturation. The manuscript investigates an important food process associated with prolonged shelf-life and the development of complex sensory profiles. The manuscript has several points that need to be addressed. 

·      There is no justification for the proposed study in the abstract section. The part written in the intro is not very convincing or satisfying. Why this study was conducted should have been elaborated better.

·       The use of molasses and its content should be mentioned in the manuscript. 

·      Line 58- 59: This sentence should be rewritten to improve clarity.

·      Line 87- “in the meanings of” should be removed, and the sentence should be revised. 

·      The introduction should include information related to the parameters of this study and the implications of the changes in these parameters (i.e., thermal denaturation temperature, enthalpy, etc.)  

·      The materials and methods part could benefit from revision. Please move the materials used in the study to a subsection called “materials” and list the ingredients, together with the relevant information (purchase, analytical grade, etc.). The methods section should be another subsection where a proper explanation of the analytical techniques is given. 

·      Line 131: Change the title of this subsection to “Preparation and thermal analysis of samples

·      In the 3.1 section,  proximate analysis results should be given, including the error +-

·      Table 1, line 224 has a typo error “treatman”

·      Do the authors think Table 1 results were discussed adequately considering the statistical outcomes? For example, for Tdo1 and Tdp1, heating rates 1.5 to 10 are not statistically different but only different from 0.5. Why? 

·      The use of “slight decrease” or “slight increase” should be avoided if there is no statistical difference among the data. 

·      In the discussion part (line 290), the first three paragraphs have nothing to do with the discussion. They can be moved to the intro as they are background/general knowledge related to thermal analysis, etc. 

·      The discussion part should be improved by finding previous studies on muscle meat products instead of whey proteins. 

·      Line 438: Please explain “ physiologically relevant time-scale”; it is unclear. Also, it does not sound relevant. 

Wrong use of words, typo errors, and the long sentences should be avoided. 

Author Response

Author’s answers to Review 1

  • There is no justification for the proposed study in the abstract section. The part written in the intro is not very convincing or satisfying. Why this study was conducted should have been elaborated better.

Answers:

-in the abstract section justification for proposed study is added (line21-25):

“Knowledge of the nature of meat protein denaturation of each kind of meat product is one of the necessary tools for developing the technology of meat product processing and to achieve desired quality and nutritional value. The kinetic analysis of meat protein denaturation is appropriate because protein denaturation gives rise to changes in meat texture during processing and directly affect the quality of product.”

 -The Introduction has been rewritten with emphasizes on clarification on why this study was conducted (Line 98-107):

“This study presented the formation of a new protein matrix in dehydrated meat through the interactions between molasses components and meat proteins during osmotic dehydration. These interactions involve complex processes that cause changes in the structure of meat proteins, leading to the development of a dry meat protein which is enriched with molasses components and thus can be less susceptible to microbiological spoilage and at the same time enriched with high-value nutrients. In the literature there are some data concerning of the thermal properties of fresh meat proteins [10–12] but not as much about thermal properties of dehydrated meat proteins [13]. Furthermore, there is a lack of literature data on the thermal properties of osmotically de-hydrated pork meat proteins and data about the kinetic behaviour of osmotically de-hydrated (in sugar beet molasses) pork meat proteins.”

And lines 125-132:

“In this study, kinetic properties, and thermal characteristics of fresh and osmotically dehydrated pork meat (Longissimus dorsi) proteins were followed by two kinetical models: iso conversional differential Friedman method [14–16] and an integral method for isoconversional data by Ortega [17]. Obtained kinetic triplet (Ea- activation energy, A pre-exponential factor and f (α) extent of conversion) has been discussed. The aim of this work is to thermally characterize (by means of differential scanning calorimetry, DSC), and compare the fresh and dried meat proteins denaturation by obtaining the kinetic and thermodynamic parameters of denaturation process.”

  • The use of molasses and its content should be mentioned in the manuscript.

 Answer

In recent years, the possibilities of using sugar beet molasses as an osmotic solution have been investigated, which gave good results in terms of the technological parameters of the process (dry matter content, water loss and increase in dry matter) and the nutritional characteristics of the obtained semi-products. ( Filipović et al., 2012, Šarić et al,, Nićetinet al.,)

The use of molasses was mentioned in the section 1. Introduction lines 50-63:

“Based on numerous studies, sugar beet molasses, a concentrated multicomponent system that is a byproduct of the sugar industry, can be used as a highly suitable hypertonic solution [6]. Molasses has a very complex chemical composition. It is known that over 200 different compounds are present in molasses [6]. It contains over 80% dry matter, around 50% sucrose, 1% raffinose, 0.25% invert sugars, around 5% protein, around 1.5% purine and pyrimidine bases, some organic acids, and pectin. It contains B-group vitamins and minerals, with significant amounts of potassium, sodium, and iron [6,7]. Due to various phytochemical compounds, including phenols, plant sterols, flavonoids and policosanols, molasses have gained interest due to their antioxidant activity, cholesterol-lowering properties, and other potential health benefits [8]. The composition of molasses is variable and primarily depends on the compounds present in the raw sugar beet, composition of molasses also depends on the way of carrying out the technological processes of sugar beet cleaning and the sugar crystallization process. [6].”

And in the section 2. Material and Methods :2.2 Osmotic Dehydration of pork meat in Sugar Beet Molasses (lines 145-152):

“Osmotic dehydration of fresh pork meat (Longissimus dorsi) was performed using sugar beet molasses. Fresh pork meat (Longissimus dorsi), 48 h post-mortem, were cleansed from external fat and connective tissue and imparted to small pieces (1×1×1) cm, then osmotic dehydration was performed using sugar beet molasses). The average dry matter content of molasses, determined by refractometry, was 84.54%. To achieve the desired concentrations of the osmotic solution, molasses was diluted with distilled water. Dehydration was performed at a temperature of 22 °C for 5 hours. The ratio of osmotic solution to meat was 5:1 (w/w).”

References

Filipović, V., Ćurčić, B., Nićetin, M., Plavšić, D., Koprivica, G.M. (2012). Mass transfer and microbiological profile of pork meat dehydrated in two different osmotic solutions. Hemijska industrija, 66 (5), 743-748

Ljubiša Šarić et al., Sugar beet molasses: properties and applications in osmotic dehydration of fruits and vegetables, Food and Feed Research, 43 (2), 135-144, 2016

MR Nićetin, LL Pezo, BL Lončar, V Filipović, DZ Šuput The possibility of increasing the antioxidant activity of celery root during osmotic treatment, Journal of the Serbian Chemical Society 82 (3), 253-2652017

  • Line 58- 59: This sentence should be rewritten to improve clarity.

Answer

The sentence was rewritten to improve clarity. Now the sentence is: (lines 60-63)

“The composition of molasses is variable and primarily depends on the compounds present in the raw sugar beet, composition of molasses also depends on the way of carrying out the technological processes of sugar beet cleaning and the sugar crystallization process.”

  • Line 87- “in the meanings of” should be removed, and the sentence should be revised.

Answer

The sentence was revised. Now sentence is: (lines 129-132)

“The aim of this work is to thermally characterize (by means of differential scanning calorimetry, DSC), and compare the fresh and dried meat proteins denaturation by obtaining the kinetic and thermodynamic parameters of denaturation process.”

  • The introduction should include information related to the parameters of this study and the implications of the changes in these parameters (i.e., thermal denaturation temperature, enthalpy, etc.)

 Answer

The obtained parameters of the proteins thermal denaturation as are compared to literature data in the Discussion, in the Introduction is mentioned that denaturation parameters are crucial for maintenance of quality of product and to obtain technological processes which are appropriate to keeping best nutritional properties of the meat food product.

  • The materials and methods part could benefit from revision. Please move the materials used in the study to a subsection called “materials” and list the ingredients, together with the relevant information (purchase, analytical grade, etc.). The methods section should be another subsection where a proper explanation of the analytical techniques is given.

Answer

Authors are grateful to this comment. The Materials and method section was revised according to referee’s instructions.

  • Line 131: Change the title of this subsection to “Preparation and thermal analysis of samples

Answer

The subsection title was not changed according with referee’s instruction, as authors considered that the suggestion was not appropriate.

  • In the 3.1 section, proximate analysis results should be given, including the error +-

Answer

 In the section 3.1. Proximate composition of pork meat (Longissimus dorsi) and sugar beet molasses

, the results were given with error (+-) included.

  • Table 1, line 224 has a typo error “treatman”

Answer:

Thank you for pointing out the typing error. It has been corrected throughout the entire text.

  • Do the authors think Table 1 results were discussed adequately considering the statistical outcomes? For example, for Tdo1 and Tdp1, heating rates 1.5 to 10 are not statistically different but only different from 0.5. Why?

Answer

Different uppercase letters within the same column and same Heating rate indicate a significant difference of means. The letters do not refer to different heating rates. The aim was to demonstrate that the drying process of meat leads to a significant shift in Tdo and Tdp at all heating rates.

  • The use of “slight decrease” or “slight increase” should be avoided if there is no statistical difference among the data.

Answer

Use of “slight decrease” or “slight increase” is avoided if there is no statistical difference among the data. The sentence is changed to (lines 349-351) : “It has been shown that activation energies obtained for fresh meat protein P1 increased with extent of conversion and for protein P2 and P3 remain about value (P2) (Fig.2a), with no statistical significance. “

  • In the discussion part (line 290), the first three paragraphs have nothing to do with the discussion. They can be moved to the intro as they are background/general knowledge related to thermal analysis, etc.

Answer: The first three paragraphs from the Discussion were moved and incorporated to the Intro.

  • The discussion part should be improved by finding previous studies on muscle meat products instead of whey proteins.

Answer:

Authors fully agree with this statement but unfortunately there no literature data about denaturation of dried meat proteins, so authors find appropriate to compare results of this work to the studies of highly concentrated whey proteins.

  • Line 438: Please explain “physiologically relevant time-scale”; it is unclear. Also, it does not sound relevant.

Answer:

The sentence was corrected, and now is:

“Therefore, as it has been stated by Sanches-Ruiz [12] biological function requires kinetic stabilization, i.e., a significant free-energy barrier along the way from N to F.” (line 450-452)

Comments on the Quality of English Language

Wrong use of words, typo errors, and the long sentences should be avoided.

Answer:

As kindly suggested, the manuscript was checked by professional English-speaking person.

Reviewer 2 Report

In my opinion the article needs some improvements.

Some examples of improvements are presented below:

-          Please use in the Introduction a smaller number of paragraphs, as well as the Materials and methods (subsection 2.1.) Please add more references in the Introduction.

-          I suggest to remove the space before [5] (Row 116).

-          Row 128: Please explain the acronym of multilayer polymer structure PVC / PE-EVOH-PE.

-          Row 128: For Differential scanning calorimetric, please use the acronym.

-          Please present the units of measure without a fraction line (kJ/mol, e.g. kJžmol-1). Please look for this aspect in the whole article.

-          Row 150: Please explain the acronym Al.

-          Row 186: Please add comma after 37.7%

Author Response

Author’s answers to Review 2

-          Please use in the Introduction a smaller number of paragraphs, as well as the Materials and methods (subsection 2.1.) Please add more references in the Introduction

ANSWER:

Introduction was rearranged and some new references has been added, also the number of paragraphs was decreased, also and in the Materials and methods number of paragraphs was lowered.  

-          I suggest to remove the space before [5] (Row 116).

Answer

The space has been removed.

-          Row 128: Please explain the acronym of multilayer polymer structure PVC / PE-EVOH-PE.

Answer

The explanation of acronym was added (PVC / PE-EVOH-PE: Polyvinyl Chloride/ Polyethylene -Ethylene-vinyl alcohol- Polyethylene) Line 155-156.

-          Row 128: For Differential scanning calorimetric, please use the acronym.

Answer

The correction of Differential scanning calorimetric, the acronym was used. Line 168

-          Please present the units of measure without a fraction line (kJ/mol, e.g. kJžmol-1). Please look for this aspect in the whole article.

Answer

The units of measure were changed and are presented without fraction line

-          Row 150: Please explain the acronym Al.

Answer

This is mistake, typo, it was corrected into aluminium. Row 185

-          Row 186: Please add comma after 37.7%

Answer

 The coma was added.

Reviewer 3 Report

Thermal characterisation and kinetic analysis of osmotically dried pork meat Longissimus dorsi

This manuscript opens the door for unusual drying method dipping fresh port meat in molasses. Authors studied thermal properties of fresh and osmotically dried material by DSC and detected the changes caused by thermal denaturation of different proteins: myosin, collagen and acting.

I think that such study can help to understand the structure of osmotically dried port meat and kinetics of changes.

I have only minor comments remarks on small mistakes in the text given in the following list.

-          Line 66 I hope that it would be good to cite Lončar et al [6].

-          Line 116, please remove year 2013.

-          Lines 123 and 130 declare the same source of molasses. Remove on line 130 words describing molasses origin.

-          Equation (2): on the left side there is missing symbol “Da”. I hope that equation (2) is discretized version of the equation (1) and should it follow exactly.

-          Part 3.1 does not contain any process how the osmotically dried meat was separated from molasses. How the meat was de-contaminated?

-    Line 224 and many other lines (e.g. text below Table 2) describing statistical procedure ANOVA: there is used the word “treatman” instead “treatment”. Treatman is person that takes care (for example person in hospital or so) but “treatment” means to do some processing.

-          Line 235 true family name of author is “Istrate”.

-          Lines 381-382 contain the sentence about rate of protein denaturation as dependent on moisture content. I fully agree with sentence that low moisture content in protein is able almost stop denaturation. My own experience with dried egg white. There was task to kill Salmonella and sporeforming microorganisms. When we increased temperature above 82°C system generated insoluble powder. In case that we wait until water content was below 2 % we were able to increase temperature up to 120 °C and keep solubility.

-          Line 515 “pH” not “PH”.

-          Line 577 author Chiou has name Bo-Sen. It can hardly be only “Sen”.

Author Response

Author’s answers to Review 3

-          Line 66 I hope that it would be good to cite Lončar et al [6].

Answer

The research of Loncar was cited, the sentence is:

“Research of Lončar et al [6] has shown that sugar beet molasses is an extremely effective medium for osmotic dehydration of fruits, vegetables, and meat.” Line 69-70

-          Line 116, please remove year 2013.

Answer

The year was deleted. Now the sentence is:

“The osmotic dehydration procedure was performed at the Faculty of Technology, University of Novi Sad, according to the procedure described in the work Šuput et al [5].” Line 140-141

-          Lines 123 and 130 declare the same source of molasses. Remove on line 130 words describing molasses origin.

Answer

The words describing the molasses origin was deleted, as it was requested.

-          Equation (2): on the left side there is missing symbol “Da”. I hope that equation (2) is discretized version of the equation (1) and should it follow exactly. 

Answer

The equations are correctly written. Equation 1 refers to the Friedman differential method, while Equation 2 pertains to the Ortega integral method. For more information, please refer to the cited references.

-          Part 3.1 does not contain any process how the osmotically dried meat was separated from molasses. How the meat was de-contaminated?

 Answer

 The sentence was added in the Materials and methods section 2.2

After the processing time, the meat samples were lightly washed with distilled water and gently blotted to remove excessive water from the surface. Line: 152-154

-    Line 224 and many other lines (e.g. text below Table 2) describing statistical procedure ANOVA: there is used the word “treatman” instead “treatment”. Treatman is person that takes care (for example person in hospital or so) but “treatment” means to do some processing.

Answer:

Thank you for pointing out the typing error. It has been corrected throughout the entire text.

-          Line 235 true family name of author is “Istrate”.

ANSWER

The author family name is corrected to Istrate. Line 269

-          Lines 381-382 contain the sentence about rate of protein denaturation as dependent on moisture content. I fully agree with sentence that low moisture content in protein is able almost stop denaturation. My own experience with dried egg white. There was task to kill Salmonella and sporeforming microorganisms. When we increased temperature above 82°C system generated insoluble powder. In case that we wait until water content was below 2 % we were able to increase temperature up to 120 °C and keep solubility.

Answer

Authors are sincerely satisfied, since this statement also confirms their results obtained in this work.

 -          Line 515 “pH” not “PH”.

Answer

The mistake was corrected, the “pH” is inserted. Line 540

-          Line 577 author Chiou has name Bo-Sen. It can hardly be only “Sen”.

Answer

The authors apologize for this gross error, the cited author name has been corrected. Authors are grateful for this comment. (Reference no 41 in manuscript)

Reviewer 4 Report

The manuscript discussed a study on the DSC and thermal denaturing kinetic study of osmotically dehydrated pork meat. The topic is interesting. However, there are still some concerns and questions that the authors need to address.

1. The English writing of this manuscript needs significant improvement, please refer to a professional English editing service to fix the grammar errors, tense, and improve the readability.

2. Please make them clear whether the authors studied pork meat or meat protein from the beginning and be consistent throughput the paper, including the title.

3. Please clearly state what kind of kinetic was studied from the beginning of the paper. Kinetic study is a very general term.

4. Line 19: what is dried meat protein? protein is the same protein as it was before the drying, they experienced destabilization, denaturing or other conformational changes during the drying.

5. Line 191: for dehydrated pork with 0.729 water activity, was it considered safe from microbial contamination or spoilage? was it too high?

6. Line 291-315: These two paragraphs are not discussions of the results obtained from this study. Instead, they are just general introduction of the concept of protein denaturing.

7. The authors explained that dried meat protein was more thermally stable by referring to literatures and DSC results. However, these are just assumptions. The authors need to provide more direct experimental data to support and justify their conclusions. It is highly recommended to perform FTIR or other approaches to directly characterize any protein structural changes or denaturing. Otherwise, the study is somewhat simple.

8. In addition, it is suggested that the author should emphasize the key value and significance of the findings from this study. Currently, a large part of the contents in the conclusion are still assumptions and postulations.

Needs significant improvement.

Author Response

Author’s answers to Review 4

  1. The English writing of this manuscript needs significant improvement, please refer to a professional English editing service to fix the grammar errors, tense, and improve the readability.

Answer

The English has been referred to the professional English speaking person.

  1. Please make them clear whether the authors studied pork meat or meat protein from the beginning and be consistent throughput the paper, including the title.

 Answer

Thank you for this comment. In this work the subject of study were pork meat proteins. The changes were made from the beginning to the end of the paper including the title.

  1. Please clearly state what kind of kinetic was studied from the beginning of the paper. Kinetic study is a very general term.

Answer

The authors are grateful for a useful comment related to the kinetic analysis.

In the title the kind of used kinetical study was added. In the Abstract first sentences it is stated that kinetic of meat protein denaturation was followed by two isoconversional kinetical methods: differential – Friedman method and an integral – Ortega method.(“Kinetic properties and thermal characteristics, obtained by differential scanning calorimetry of fresh and osmotically dehydrated pork meat proteins (Longissimus dorsi) proteins were followed by two isoconversional kinetical methods: a differential – Friedman and an integral – Ortega method.”)

In the Introduction it was described that: “In this study, kinetic properties and thermal characteristics of fresh and osmotically dehydrated pork meat (Longissimus dorsi) proteins were followed by two kinetical models: iso conversional differential Friedman method [14–16] and an integral method for isoconversional data by Ortega [17]. Obtained kinetic triplet (E- activation energy, A- pre-exponential factor and f (α) extent of conversion) has been discussed.”

In the section “2. Materials and methods” it was presented in more detail, upon the title “Kinetics of meat protein thermal denaturation” what kind of kinetic methods has been employed:

“The activation energy (E) of the protein denaturation process in both fresh and dried meat samples were determined using two isoconversional non-isothermal methods. The first method applied was the differential method developed by Friedman (Eq. (1)) [16], while the second method employed was the integral method developed by Ortega (Eq. (2)) [17].”

In the Result and in the Discussion the obtained results of two kinetical methods of meat protein denaturation were presented and discussed.

Also, in the title the method of the kinetic analysis was added, and now tittle is:

Thermal characterisation and isoconversional kinetic analysis of osmotically dried pork meat proteins Longissimus dorsi.

Authors sincerely consider that this answer is satisfactorily.

  1. Line 19: what is dried meat protein? protein is the same protein as it was before the drying, they experienced destabilization, denaturing or other conformational changes during the drying.

Answer

Authors highly agree with referee statement, in the addition the authors would also emphasize that the same protein was also changed due to interaction with the components of the osmotic solution, since it interacted with these components during osmotic dehydration, which led to the creation of characteristic features of dried meat proteins, which are the subject of this study.

  1. Line 191: for dehydrated pork with 0.729 water activity, was it considered safe from microbial contamination or spoilage? was it too high?

 Answer

The water activity (aw) value obtained for osmotically dehydrated meat proteins, as it is well known, is in not considered as safe regarding microbial spoilage as values below aw 0.6 were considered as safe considering microbial contamination. Šuput et al. (Šuput et al 2013) determined the stability of osmotically dehydrated meat in sugar beet molasses, in terms of microbiological stability - total microorganisms (TVC) and Enteroabcteriaceae, physicochemical properties such as pH, water activity, moisture, amount of fats, carbohydrates, oxidative stability (measured as thiobarbituric acid TBA). Suput et al concluded that samples remained microbiologically stable during the storage period of 60 dayes.

Reference:

Danijela Šuput, Vera Lazić, Lato Pezo, Jasmina Gubić, Branislav Šojić, Dragana Plavšić, Biljana Lončar, Milica Nićetin, Vladimir Filipović, Violeta Knežević Shelf life and quality of dehydrated meat packed in edible coating under modified atmosphere, Rom Biotechnol Lett. 2019; 24(3): 545-553.

doi: 10.25083/rbl/24.3/545.553

  1. Line 291-315: These two paragraphs are not discussions of the results obtained from this study. Instead, they are just general introduction of the concept of protein denaturing.

Answer

Authors agree with this statement, those paragraphs were moved and incorporated into Introduction.

  1. The authors explained that dried meat protein was more thermally stable by referring to literatures and DSC results. However, these are just assumptions. The authors need to provide more direct experimental data to support and justify their conclusions. It is highly recommended to perform FTIR or other approaches to directly characterize any protein structural changes or denaturing. Otherwise, the study is somewhat simple.

Answer

Thank you for this comment. The author’s goal was to demonstrate that with of DSC technique is possible to easily describe complex processes which take place during osmotic dehydration of meat proteins, and that the DSC technique can be an excellent tool for following the changes that occur during the process of dehydration. Authors, would like to emphasize that it has been shown by Wortmann et al that “the DSC yields the denaturation enthalpy ΔHD which depend on the amount and structural integrity of the α-helical material in the intermediate filaments (IF), and the temperature TD which is kinetically controlled by the cross-link density of the matrix (IFAPs) in which the IFs are embedded. “Meaning that changes in protein secondary structure affect the DSC protein denaturation curve, DSC result. Also, in the study of Tuan Phan-Xuan et al it was found that differences in protein secondary structure affect DSC results.  The authors agree that the usage of the FTIR, as referee suggested, or Raman spectroscopy should be performed, but in further study of osmotically dried proteins.

References

Wortmann, F. J.; Springob, C.; Sendelbach, G. J. Investioations of cosmetically treated human hair by differential scanning calorimetry in water, Cosmet. Sci. 2002, 53, 219-228.

Tuan Phan-Xuan, Ekaterina Bogdanova, Jens Sommertune, Anna Millqvist Fureby, Jonas Fransson, Ann E. Terry, Vitaly Kocherbitov, The role of water in the reversibility of thermal denaturation of lysozyme in solid and liquid states Biochemistry and Biophysics Reports 28 (2021) 101184

  1. In addition, it is suggested that the author should emphasize the key value and significance of the findings from this study. Currently, a large part of the contents in the conclusion are still assumptions and postulations.

Answer

The Conclusion was redesigned as referee proposed. The authors emphasized the in the conclusion the key values and significance of the results from this study.

“The achieved thermal stabilization, through protein-protein association and interactions with the osmotic solution (molasses), resulted in a dry meat protein enriched with valuable nutritional compounds. It was proposed that dried meat proteins are in partly unfolded state and regarding the kinetic stability follows a scenario “one” de-scribed previously. Therefore, it can be concluded that the osmotically dried meat proteins are thermally stabilized but thermodynamically and kinetically less stable in comparison to the fresh meat proteins.” (Lines 478-484)

 The sentence was added (lines 489-492):

“It was demonstrated that with simple usage of DSC technique is possible to describe complex processes which take place during osmotic dehydration of meat proteins, and that the DSC technique can be an excellent tool for following the changes that occur during the process of dehydration.”

Comments on the Quality of English Language

Needs significant improvement.

Answer

As kindly suggested, the manuscript was checked by professional English-speaking person.

Round 2

Reviewer 1 Report

Some of the comments suggested by the Referee were addressed. The current form is okay except two major and a few minor things. 

1. Line 29: Instead of "the process of removing", "a process of removing" 

2. Major: Line 40-41: This sentence must be revised as drying process inhibits the growth of microorganisms. 

3. Line 35: instead "human food", "human nutrition" 

4. Major: As the authors utilized molasses, which contains sugar,  in their study yet no sugar-protein interaction discussion is included which leaves the manuscript content as insufficient. 

5. Line 479-480: "enriched with valuable nutritional compounds" must be removed as the authors did not quantify the nutritional enrichment caused by the soaking the meat in molasses solution. This cannot be concluded given the method does not involve the corresponding nutrition quantification study. 

Still requires minor editing. 

Author Response

  1. Line 29: Instead of "the process of removing", "a process of removing"

Answer:

On line 29 correction was done into "a process of removing"

  1. Major: Line 40-41: This sentence must be revised as drying process inhibits the growth of microorganisms.

Answer:

Authors are sincerely grateful for this reviewer’s notice!  Authors fully agree with the comment, as sentence was completely inappropriate. Sentence was revised, and now sentence is (line 40-41):

Reducing the water content in meat helps in preventing microbial growth, which in turn prevents the presence of harmful substances in meat (4).

 (Reference 4: Mediani, A.; Hamezah, H.S.; Jam, F.A.; Mahadi, N.F.; Chan, S.X.Y.; Rohani, E.R.; Che Lah, N.H.; Azlan, U.K.; Khairul Annuar, N.A.; Azman, N.A.F.; et al. A Comprehensive Review of Drying Meat Products and the Associated Effects and Changes. Front Nutr 2022, 9, 1057366, doi:10.3389/FNUT.2022.1057366/BIBTEX.)

  1. Line 35: instead "human food", "human nutrition"

Answer

The correction has been made, line 36: “nutrition“ instead ”human” is replaced.

  1. Major: As the authors utilized molasses, which contains sugar, in their study yet no sugar-protein interaction discussion is included which leaves the manuscript content as insufficient.

Answer

Authors are grateful for this comment, and authors fully agree with reviewer’s comment!

The protein-sugar interaction was discussed in the section 4 Discussion:

It is well known that sugar solutions especially sucrose (which is most abundant carbohydrate present in molasses [ref 6, Sjölin 2020]) can stabilize the protein structure [Lee 1981].  By now, most studied   phenomenon which was studied was marked increase in thermal and conformational stability of the globular proteins in aqueous medium in the presence of sugar [Semenowa 2002].  The mechanism of the action of sugar on protein was proposed to be that the sugar molecules are preferentially excluded from the region immediately surrounding the proteins (Semenowa 2002).  Ferreira et al. (Ferreira 2019) studied the effects of osmolytes (sucrose and trehalose) on protein–solvent interactions in crowded environments and found that protein’s responses to the presence of different osmolytes are governed by the protein’s structures. Cao et al. (Cao 2022) gave the systematic analysis of protein–carbohydrate inter-actions and binding sites. Those findings, considered the globular protein stabilization in aqueous medium by sugar molecules, can be applied to this work as well, since meat fibrillar proteins clearly interact with the osmotic solution (sugar cane molasses) and the dried proteins are thermally stabilized and partly unfolded (Table 1).

References:

James C Lee and Serge N Timasheff, The stabilization of proteins by sucrose, J Biol. Chem. 256,7193-7201.

Luisa A. Ferreira,a Olga Fedotoff,a Vladimir N. Uversky*bcde and Boris Y. Zaslavsky*a Effects of osmolytes on protein–solvent interactions in crowded environments: study of sucrose and trehalose effects on different proteins by solvent interaction analysis

Yiwei Cao, Sang-Jun Park, and Wonpil Im A systematic analysis of protein–carbohydrate interactions in the Protein Data Bank, Glycobiology, 2021, vol. 31, no. 2, 126–136, doi: 10.1093/glycob/cwaa062

  1. Line 479-480: "enriched with valuable nutritional compounds" must be removed as the authors did not quantify the nutritional enrichment caused by the soaking the meat in molasses solution. This cannot be concluded given the method does not involve the corresponding nutrition quantification study.

Answer

Authors are thankful for this assumption and completely agree. The sentence was corrected, "enriched with valuable nutritional compounds" was deleted and now sentence is (line 495-497):

“The achieved thermal stabilization, through protein-protein association and interactions with the osmotic solution (molasses), resulted in the formation of dry meat protein.”

Comments on the Quality of English Language

Still requires minor editing.

Answer:

The quality of English Language was checked by professional English-speaking person.

Reviewer 4 Report

The authors have addressed my concerns and questions.

No

Author Response

The authors express gratitude for the valuable suggestions provided by the reviewer, as they significantly contributed to the improvement of the manuscript.